# Biocompatible Ir(III) Complexes as Oxygen Sensors for Phosphorescence Lifetime Imaging

**DOI:** 10.3390/molecules26102898

**Published:** 2021-05-13

**Authors:** Ilya S. Kritchenkov, Anastasia I. Solomatina, Daria O. Kozina, Vitaly V. Porsev, Victor V. Sokolov, Marina V. Shirmanova, Maria M. Lukina, Anastasia D. Komarova, Vladislav I. Shcheslavskiy, Tatiana N. Belyaeva, Ilia K. Litvinov, Anna V. Salova, Elena S. Kornilova, Daniel V. Kachkin, Sergey P. Tunik

**Affiliations:** 1Institute of Chemistry, St. Petersburg State University, Universitetskii av., 26, 198504 St. Petersburg, Russia; i.s.kritchenkov@spbu.ru (I.S.K.); nastisol@gmail.com (A.I.S.); st055671@student.spbu.ru (D.O.K.); v.porsev@spbu.ru (V.V.P.); v.sokolov@spbu.ru (V.V.S.); 2Institute of Experimental Oncology and Biomedical Technologies, Privolzhskiy Research Medical University, Minin and Pozharsky sq. 10/1, 603005 Nizhny Novgorod, Russia; shirmanovam@gmail.com (M.V.S.); kuznetsova.m.m@yandex.ru (M.M.L.); komarova.anastasii@gmail.com (A.D.K.); vis@becker-hickl.de (V.I.S.); 3Becker&Hickl GmbH, Nunsdorfer Ring 7-9, 12277 Berlin, Germany; 4Institute of Cytology of the Russian Academy of Sciences, Tikhoretsky av. 4, 194064 St. Petersburg, Russia; tatbelyaeva@gmail.com (T.N.B.); lik314@mail.ru (I.K.L.); avsalova@gmail.com (A.V.S.); lenkor@incras.ru (E.S.K.); 5Institute of Biomedical Systems and Biotechnology, Peter the Great St. Petersburg Polytechnical University, Khlopina Str. 11, 194021 St. Petersburg, Russia; 6Faculty of Biology, St. Petersburg State University, Universitetskaya emb., 7/9, 199034 St. Petersburg, Russia; pspdaniel@mail.ru

**Keywords:** Ir(III) complexes, NIR emitters, phosphorescence lifetime imaging, oxygen sensing

## Abstract

Synthesis of biocompatible near infrared phosphorescent complexes and their application in bioimaging as triplet oxygen sensors in live systems are still challenging areas of organometallic chemistry. We have designed and synthetized four novel iridium [Ir(N^C)_2_(N^N)]^+^ complexes (N^C–benzothienyl-phenanthridine based cyclometalated ligand; N^N–pyridin-phenanthroimidazol diimine chelate), decorated with oligo(ethylene glycol) groups to impart these emitters’ solubility in aqueous media, biocompatibility, and to shield them from interaction with bio-environment. These substances were fully characterized using NMR spectroscopy and ESI mass-spectrometry. The complexes exhibited excitation close to the biological “window of transparency”, NIR emission at 730 nm, and quantum yields up to 12% in water. The compounds with higher degree of the chromophore shielding possess low toxicity, bleaching stability, absence of sensitivity to variations of pH, serum, and complex concentrations. The properties of these probes as oxygen sensors for biological systems have been studied by using phosphorescence lifetime imaging experiments in different cell cultures. The results showed essential lifetime response onto variations in oxygen concentration (2.0–2.3 μs under normoxia and 2.8–3.0 μs under hypoxia conditions) in complete agreement with the calibration curves obtained “in cuvette”. The data obtained indicate that these emitters can be used as semi-quantitative oxygen sensors in biological systems.

## 1. Introduction

Biochemical processes in living aerobic organisms in many aspects are critically dependent on oxygen, which is a key substrate of energy metabolism [1] and an important indicator of cell/tissue physiological status under healthy and diseased conditions. Misbalance of oxygen demand and supply is associated with many pathologies, such as cancers [2,3,4], neurological diseases [5,6], retinopathy, glaucoma, and nuclear cataracts [7,8,9], that inevitably stimulates the study of organs, tissues, and cell compartments’ oxygenation, see for example Chapters 4 and 12 in book [10], Chapter 17 in [11], and research articles [12,13,14]. Noteworthy that in vitro and in vivo real time monitoring of O_2_ concentration with high spatial and temporal resolution is one of the challenging tasks in medicine and biology, the solution of which can provide invaluable information related to metabolic status of cells and tissues in normal and diseased states. Among various modes of oxygen mapping in biological systems phosphorescence lifetime imaging (PLIM) provides considerable advantages due to its noninvasive nature, and reversible and rapid response onto variations in local oxygen content, which can be detected both at qualitative and quantitative levels either showing relative changes or giving absolute values of the analyte concentration. PLIM method is based on the phenomenon of the phosphorescence dynamic quenching by molecular oxygen due to excitation energy transfer from triplet excited state of emitter to O_2_ [15,16], which is a unique small molecule with triplet ground state, that ensure extremely high selectivity of this sensing procedure. The instrumentation for the PLIM measurements is now commercially available and data processing algorithms are very well developed [17,18]. Qualitative assessments of the hypoxia regions in living organisms and cell compartments have been already performed by using a range of Pt, Pd, Ir, and Ru phosphorescent complexes, see reviews [16,19,20,21,22] and some recent experimental papers [23,24,25,26,27,28,29], which demonstrated different lifetime sensitivity to oxygen and diverse levels of biocompatibility/toxicity. However, to exploit in full the PLIM mode of oxygen sensing, in particular, in order to obtain the quantitative spatial and temporal distribution of O_2_ in biological systems, the targeted chemical design of the sensor composition and structure is highly necessary.

The oxygen sensor works in a complicated physiological media containing a large variety of biomolecules, which may absorb relatively small and often hydrophobic sensor molecules thus changing Stern–Volmer quenching constant in an unpredictable manner. To prevent the cross-talking of several orthogonal parameters, which can significantly distort the result of the target value (O_2_ concentration) measurement, the sensor needs protection from the interaction with the other components of physiological media. Two research groups from Pennsylvania and Cork universities successfully solved this problem by packing the chromophore center(s) either into dendrimeric polyethyleneglycol (PEG) ligand corona [30,31,32] or into polymeric nanospecies [33,34,35] that prevents the chromophore pocketing into hydrophobic cavities of biomolecules. In both cases the emitters are based on platinum and palladium porphyrins, which demonstrate high quantum yields (QY), wide interval of lifetime variations in response to the changes in O_2_ concentration, and emission in the red and near infrared (NIR) regions that are well suited for sensing in biological systems. However, some specific features of these sensors limit their utilization, e.g., PEGylated Pt/Pd porphyrins (Oxyphors family), require challenging synthesis and are of rather large size (ca. 7 nm [32]) which makes impossible their uptake by cells and implies intravascular sensing as a major area of application. The nanoparticle based sensors (NanO2 and analogous species) display excellent brightness due to incorporation of a few emitter molecules into the particle core. However, being prepared of charged hydrophobic polymers [34,35] they lose electrostatic stability at physiological ionic strength and tend to aggregate in aqueous saline [33,36]. From the viewpoint of PLIM, one of the major problems for both types of sensors is very long emission lifetime, which typically span the interval of 30–70 μsec under physiological conditions that makes difficult real-time monitoring of oxygen concentration in appreciable spatial areas in microscopic experiments.

These considerations prompted us to investigate another popular luminescent platform [37,38] based on bis-cyclometalated iridium complexes with the aim to obtain water-soluble compounds suitable for effective application in biological systems as quantitative oxygen sensors in PLIM mode. These phosphorescent complexes (a) are relatively easy to prepare; (b) demonstrate high versatility relative to the structure and composition of ligand environment; (c) consequently allow for wide variations in photophysical parameters of emitter and chemical/biological properties of the target molecules, (d) are bright and stable to photobleaching luminophores. The last but not the least, emission lifetime changes through normal to hypoxic conditions fall in the range from hundreds of nanoseconds to a few microseconds that is more than order of magnitude shorter compared to the platinum and palladium porphyrins mentioned above. In our recent publication [27] we presented a series of iridium [Ir(N^C)_2_(N^N)]^+^ complexes, which contain oligo(ethyleneglycol) (OEG) functions {–C(O)NHCH-(CH_2_OC_2_H_4_OC_2_H_4_OCH_3_)_2_} at the periphery of the N^N and/or N^C ligands that made the hydrophobic core of these molecules water-soluble and suitable for semi-quantitative PLIM experiments on cell cultures. It has to be noted that the poly- and oligo-ethyleneglycole pendants have been already used to make molecular transition metal chromophores (Pt(II) [39,40,41], Pd(II) [42,43], Au(III) [44], Re(I) [45,46], Ir(III) [45,47,48,49], Ru(II) [50]) water-soluble, less toxic, and biocompatible. However, in all but one of these studies the authors functionalized only one of the ligands in metal coordination environment (using either OEG or PEG) that is chemically simpler but is evidently less effective compared to introduction of this pendants into all three chelates of typical iridium [Ir(N^C)_2_(N^N)]^+^ emitters. In the present article, we describe the synthesis, characterization, and photophysical study of iridium complexes functionalized at N^N and N^C ligands with branched “second generation” OEG substituents aimed at increased protection of the chromophores from the interaction with biomolecules. We compared the solubility and phosphorescence lifetime characteristics of the complexes of the “first” and “second generation” in different media to evaluate their potential as quantitative oxygen sensors in the cell studies using PLIM. We also complemented these investigations by the study the complexes’ cytotoxicity, dynamics of intracellular uptake, and their localization in three cell types.

## 2. Results

Synthesis of N^C and N^N ligands containing OEG pendants of various branching are shown in Scheme 1.

Details of experimental procedures are given in Experimental section, their ^1^H, ^1^H-^1^H COSY and NOESY NMR spectra (Appendix A) and ESI^+^ mass spectra (Appendix A) are presented in Appendix A.

The iridium complexes have been obtained by using a standard reaction sequence (Scheme 2), which consists of preparation of iridium orthometalated dimer followed by its reaction with the diimine ligand to afford the target mononuclear [Ir(N^C)_2_(N^N)]^+^ compounds. **Ir1** and **Ir2** complexes differ in the composition of the OEG pendants, whereas **Ir#** and **Ir#a** contain Cl^−^ and PF_6_^−^ counterions, respectively. The obtained compounds are well soluble and stable in aqueous solutions including model physiological media (fetal bovine serum, FBS).

The complexes were obtained as dark red amorphous solids and did not give crystals suitable X-ray crystallographic study. Therefore, their composition and structure in solution were studied by using ^1^H, ^1^H-^1^H COSY and NOESY NMR spectroscopy (Appendix A) and mass-spectrometry (Appendix A). The spectroscopic data obtained are completely compatible with the structural patterns shown in Scheme 1, see for example Figure 1.

Additionally we carried out optimization of the central core structure using DFT method, which gave molecular architecture (named **Ir0**) common for the complexes of this sort, see Figure 2, with the nitrogen atoms of the N^C ligands in *trans* position at coordination octahedron. Note that for the sake of simplicity optimization was performed for the molecule where OEG pendants were changed for methyl substituents. Cartesian coordinates of the optimized structure are given in a separate file (Ir0.xyz) supplied as Appendix A, key structural characteristics are summarized in Appendix A. These data are in good agreement with the structural parameters found earlier for the closely analogous iridium complex (Appendix A) [51].

## 3. Discussion

### 3.1. Photophysical Properties

All complexes are luminescent in aqueous media to give emission in NIR region with a rather high quantum yield, up to ca. 12% in degassed solution. Absorption, excitation, and emission spectra are shown in Appendix A and Figure 3, respectively. The numerical photophysical data are summarized in Table 1.

The complexes display large Stokes shift, lifetime in microsecond domain, and emission quenching by molecular oxygen that points to the emission origin from triplet excited state, i.e., phosphorescence. According to the data of DFT calculations for closely analogous iridium complexes [52], the chromophore in the emitters of this sort is localized onto the central fragment of the molecule and observed emission can be assigned to ligand centered (LC) transition with some admixture of the ligand to ligand (LLCT) and metal to ligand (MLCT) charge transfer. The experimental data are in complete agreement with this assignment and indicate that the OEG pendants and counterions have nearly no effect onto photophysical characteristics of these complexes because of their lateral disposition relative to the chromophore center.

It is also worth noting that lifetime sensitivity of these complexes is relatively low that is typical for transition metal chromophores (e.g., iridium [27] and ruthenium [53] emitters) in aqueous media. Nevertheless, the lifetime interval of ca. 1 μsec between the values obtained in aerated and degassed solutions is large enough to estimate potential of these sensors for application as indicators of oxygen concentration variations in biological systems. 

Dependence of emission lifetime on oxygen concentration in aqueous solution, PBS, and in FBS was investigated for the **Ir2** and **Ir2a** complexes; note that **Ir1** and **Ir1a** proved to be toxic to be used as sensors in biological systems, vide infra, therefore we did not make lifetime calibration on oxygen concentration for these complexes. The data obtained for **Ir2** and **Ir2a** are summarized in Appendix A, the corresponding Stern–Volmer plots are shown in Figure 4. We found that in PBS and FBS solutions the emission decays are best-fitted using bi-exponential analysis, see Appendix A. The shorter exponent of the decay commonly displays contribution in the range 10–20%, varies in a relatively narrow interval, and does not show systematic dependence on oxygen concentration, therefore in the calibration plots we used the lifetimes corresponding to the longer exponent, τ_2_ (Appendix A). This emission parameter does not depend on pH and complex concentration (Appendix A) but shows good linearity of the Stern–Volmer plot (Figure 4) that is indicative of dynamic character of emission collisional quenching with molecular oxygen.

The calibration plots and Stern–Volmer constants obtained in pure water and PBS solutions are similar and differ substantially from those recorded in FBS. However, it is worth noting that the lifetimes obtained in these three media are nearly identical at zero oxygen concentration, see Figure 4, whereas in the presence of oxygen, the lifetime values obtained in FBS are systematically higher and K_sv_ value is substantially lower. These data indicate that the key photophysical characteristics of the chromophores (rate constants of radiative and nonradiative transitions from the triplet excited state) do not depend on the media composition, whereas the oxygen quenching rate constants are lower in the solutions containing biomolecules. These observation are completely compatible with collisional quenching mechanism of **Ir2** and **Ir2a** phosphorescence by molecular oxygen because quenching rate constant is a function of oxygen diffusion coefficient in the corresponding solution, which in turn is proportional to the media viscosity [54]. The study of τ_2_ dependence on the viscosity of aerated FBS solutions (Appendix A) clearly shows systematic lifetime (τ_2_) growth upon increase in FBS concentration, which is accompanied by related increase in solution viscosity. These data are in line with the dynamic character of the phosphorescence quenching in the model viscous media. However, the effect of viscosity onto emission lifetime is relatively small and does not exceed 10% in the interval studied, which evidently covers the values typical for the intracellular media [55,56,57]. Comparison of the lifetime data obtained in cuvette using spectrometer and those obtained in PLIM experiments (Appendix A) showed that they coincide within the margin of experimental uncertainty. The above analysis made possible to conclude that the **Ir2** and **Ir2a** molecules are well shielded from noncovalent interaction with biomolecules, which could substantially distort the oxygen concentration determination in biological systems. The observed effect of media viscosity onto emission lifetime most probably are negligible for the measurements in biological samples and we can consider the calibration obtained in FBS solution (Figure 4) as a reliable estimation of oxygen concentration in cells using PLIM. 

### 3.2. Biological Experiments

#### 3.2.1. Cytotoxicity Tests

To assess viability of different cell lines in the presence of iridium complexes we used MTT colorimetric assay, Figure 5. The results of these tests indicate that:(i)**Ir1** displays higher toxicity compared to its relatives (**Ir2** and **Ir2a**) containing branched OEG pendants;(ii)Incubation of cancer (CT26 and HeLa) and normal (CHO-K1) cells with **Ir1** at the concentrations above 25 μM reduced cell viability by >20%;(iii)For **Ir2** comparable level of toxicity was not achieved even at the concentration 150 μM;(iv)**Ir2a** is only slightly more toxic than **Ir2**.

**Figure 5 molecules-26-02898-f005:**
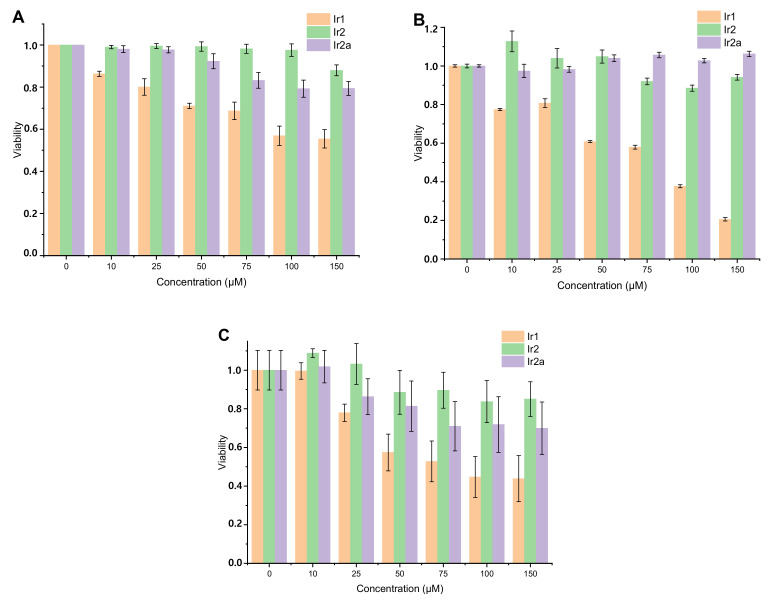
MTT assay of CT26 (**A**), HeLa (**B**), and CHO-K1 (**C**) cells after incubation with **Ir1**, **Ir2**, **Ir2a** for 24 h at different concentrations. Cell viability of control cells (without probe) was taken for 1. Mean ± standard deviation. N = 30 repetitions for CT26, 15 repetitions for HeLa, and 5 repetitions for CHO-K1 cells.

These results are not unexpected because of higher hydrophobicity/lipophilicity of **Ir1** compared the complexes of **Ir2** series that was also evidenced by lower solubility of **Ir1** in aqueous media. These observations are in agreement with the correlation between lipophilicity of metal complexes and their toxicity found earlier for PEGilated rhenium [46], platinum [58], and iridium [47,59] chromophores, which were used in biological experiments and studied from the viewpoint of their biocompatibility. These findings showed that applicability of **Ir1** in biological experiments is limited to a low concentration range that makes difficult obtaining high quality PLIM data. Therefore all further imaging experiments have been performed with **Ir2** and **Ir2a**.

#### 3.2.2. Dynamics of the Complexes’ Uptake.

In order to find out whether **Ir2** and **Ir2a** are accumulated by the living cultured cells and to study the dynamics of the process, we monitored intracellular luminescence of the complexes using confocal laser scanning microscopy; see Figure 6 and Appendix A. We found that both complexes enter different cell types and their luminescence intensity gradually increases with the time of incubation, indicating the increase of intracellular probe concentration. Maximum luminescence intensity was observed after 24-h incubation. It is important that the cells preserved typical morphology that is indicative of the absence of toxic effects, as it is seen from the bright-field microscopic images.

A very low diffuse signal of the complexes’ emissions in HeLa or CHO-K1 cells can be detected after 4 h of incubation for the both **Ir2** and **Ir2a**. Phosphorescent signal is detected in perinuclear region after incubation for more than 4 h. However, it should be taken into account that resultant phosphorescence intensity of the complexes is not only proportional to their concentration in a certain area, but also inversely proportional to O_2_ level at the same location. This means that emission intensity can be only considered as a semi-qualitative measure of the complexes’ cellular uptake. Thus, the data obtained indicate that (i) distribution of the complexes throughout the cell is not uniform and they undergo slow accumulation into some intracellular compartment, (ii) in the areas with bright phosphorescence either the concentration of the probe is high or the low level of O_2_ is maintained. A similar uptake behavior was observed for the other two cell lines, see Appendix A.

#### 3.2.3. Colocalization Experiments.

To clarify selectivity in the probe accumulation inside the cells we tried to co-localize the areas with bright luminescent signal with the markers of some cellular compartments, using vital markers for nuclei, mitochondria, and acidic endolysosomes.

Co-staining of the cells with the complexes and organelle-specific dyes showed that both **Ir2** and **Ir2a** display preferential accumulation in the lysosomes for all cell types studied with somewhat less selectivity in the case of CT 26 cell line (Figure 7, Appendix A). Considering relatively high molecular mass of the complexes, it can be reasonably suggested that they enter the cells by nonspecific fluid phase endocytosis (FPE) and finally are delivered to lysosomes. They are unlikely penetrate directly through the plasma and mitochondria membranes and slow dynamics of the dyes appearance in lysosomes perfectly fits characteristics of FPE. 

#### 3.2.4. PLIM Experiments

Next, we assessed phosphorescence lifetimes of **Ir2** and **Ir2a** in three cell lines mentioned above under normoxic and hypoxic conditions using PLIM, Figure 8, Appendix A. HeLa, CT26, and CHO-K1 were incubated with non-toxic concentration (40–100 µM) of the complexes; hypoxia was simulated by placing cover glass over cells monolayer for 2 h (equivalent to ~1–5% O_2_ [60]). The obtained data indicate that for all cell types under study the probes PLIM signal is localized in the same areas where phosphorescence was detected by confocal microscopy. In cellular environment both probes display monoexponential phosphorescence decay (χ^2^ ≤ 1.2) with the lifetime values, which fall in the τ_2_ range found for **Ir2** and **Ir2a** in aerated and degassed FBS solutions, vide supra. Under hypoxic conditions the complexes display relatively narrow lifetime distributions; the weighted averages cover the interval 2.8 to 3.0 µs that is somewhat lower (for ca. 400 ns) than the values obtained for deaerated FBS solutions, Figure 8, Appendix A. This effect can be explained either by lower viscosity of cellular media or by incomplete deoxygenation of the corresponding compartment in living cells. A certain local inhomogeneity of the lifetime map is observed; however, the scatter does not exceed 150 ns (5%).

The lifetime distribution in aerated cells is expectedly characterized by substantially wider distribution profile and stronger inhomogeneity of the lifetime map. Similar to deaerated samples the weighted averages (2.0–2.4 µs) in aerated cells are slightly lower than τ_2_ values found in aerated FBS and this difference can be analogously ascribed to the variations in media viscosity and excessive accumulation of oxygen in a certain cell compartment. This uncertainty in the assignment of the observed local lifetime variations means that careful attention should be paid to interpretation of the PLIM results from the viewpoint of local oxygen concentrations. Nevertheless, one can see that both complexes display a good dynamic range in terms of phosphorescence lifetimes: hypoxia-induced changes are ~0.9 µs, or ~50% of the initial value, showing a high potential of the novel probes as phosphorescent sensors of intracellular oxygen.

## 4. Materials and Methods

### 4.1. Synthesis of Ligands and Complexes

General comments. The ^1^H, ^1^H-^1^H COSY and NOESY (400 MHz) NMR spectra were recorded on a Bruker 400 MHz Avance; chemical shift values were referenced to the solvent residual signals. Mass spectra were recorded on a Bruker maXis HRMS-ESI-QTOF in the ESI^+^ mode. Boc-L-aspartic acid [61], 6-(benzo[b]thiophen-2-yl)phenanthridine-2-carboxylic [52] acid and 2,5,8,12,15,18-hexaoxanonadecan-10-amine [62] (**NH_2_-2OEG**) were obtained according to the published procedures. Other reagents (Merck KGaA, Darmstadt, Germany) and solvents (Vekton, St. Petersburg, Russia) were used as received without further purification. NMR and ESI-MS spectra for all substances obtained are presented in Appendix A.

Synthesis of **Boc-NH-4OEG**. Boc-l-aspartic acid (93.2 mg, 0.400 mmol), the corresponding amine (**NH_2_-2OEG**, 129.8 mg, 0.440 mmol), benzotriazolyloxytripyrrolidinophosphonium hexafluorophosphate (PyBOP, 218.4 mg, 0.420 mmol), NEt_3_ (ca. 80 mg, 0.792 mmol) and dry DMSO (1.0 mL) were placed in a 5 mL vial. The reaction mixture was stirred for 18 h at RT and thoroughly evaporated. The dried residue was dissolved in 50 mL of chloroform and sequentially washed with (i) 5 mL of brine with 0.25 g of citric acid monohydrate, (ii) 5 mL of brine with 0.1 g of NaHCO_3_ and (iii) with 5 mL of pure brine. The organic layer was dried over Na_2_SO_4_ and then evaporated to give the oily substance. It was dissolved in EtOAc/n-hexane 5:1 mixture, and purified using column chromatography (sequentially eluted with EtOAc and EtOAc/methanol 5:1 mixture). After evaporation the resulting ligand was obtained as colorless viscous oil. 271 mg, yield 86%. ^1^H NMR (CDCl_3_, 400 MHz, δ): 7.08 (br d, 1H, NH), 6.62 (br d, 1H, NH), 6.16 (br d, 1H, NH), 4.44 (br m, 1H, CH), 4.17 (br m, 2H, CH), 3.69–3.47 (m, 40H, CH_2_), 3.39 (s, 12H, CH_3_), 2.85–2.52 (m, 2H, CH_2_), 1.45 (s, 9H, CH_3_). HRMS (ESI) *m*/*z*: 810.4576 calculated for C_35_H_69_N_3_NaO_16_^+^ [M + Na]^+^, found 810.4579.

Synthesis of **NH_2_-4OEG**. Solution of **Boc-NH-4OEG** (271 mg, 0.344 mmol) in 4 mL of 1:1 mixture of CH_2_Cl_2_/CF_3_CO_2_H was stirred for 5 h at RT in a 25 mL vial. The reaction mixture was thoroughly evaporated, the dried residue was dissolved in 100 mL of chloroform and sequentially washed two times with 10 mL of brine with addition 0.15 g of NaHCO_3_ and once with 5 mL of pure brine. The organic layer was dried over Na_2_SO_4_ and then evaporated to give the oily substance. 222 mg, yield 94%. ^1^H NMR (CDCl_3_, 400 MHz, δ): 7.68 (d, J = 8.6 Hz, 1H, NH), 6.92 (br d, 1H, NH), 4.22 (br m, 3H, CH), 3.72–3.49 (m, 40H, CH_2_), 3.40 (s, 12H, CH_3_), 2.77–2.42 (m, 2H, CH_2_). HRMS (ESI) *m*/*z*: 688.4232 calculated for C_30_H_62_N_3_O_14_^+^ [M + H]^+^, found 688.4225.

Synthesis of **N^C#** ligands (**N^C1** and **N^C2**). 6-(benzo[b]thiophen-2-yl)phenanthridine-2-carboxylic acid (53.2 mg, 0.150 mmol), the corresponding amine (**NH_2_-2OEG** or **NH_2_-4OEG**, 0.165 mmol), benzotriazolyloxytripyrrolidinophosphonium hexafluorophosphate (PyBOP, 81.7 mg, 0.157 mmol), NEt_3_ (ca. 30 mg, 0.297 mmol) and dry DMSO (0.5 mL) were placed in a 5 mL vial. The reaction mixture was stirred for 18 h at RT and thoroughly evaporated. The dried residue was dissolved in 50 mL of chloroform and sequentially washed with (i) 5 mL of brine with 0.25 g of citric acid monohydrate, (ii) 5 mL of brine with 0.1 g of NaHCO_3_ and (iii) with 5 mL of pure brine. The organic layer was dried over Na_2_SO_4_ and then evaporated to give the oily substance. It was dissolved in EtOAc/n-hexane 5:1 mixture, and purified using column chromatography (eluted with EtOAc). After evaporation the resulting ligand was obtained as colorless viscous oil.

**N^C1.** 81.5 mg, yield 86%. ^1^H NMR ((CD_3_)_2_CO, 400 MHz, δ): 9.32 (s, 1H), 9.00 (d, J = 8.0 Hz, 1H), 8.84 (d, J = 8.0 Hz, 1H), 8.32 (d, J = 9.4 Hz, 1H), 8.19 (d, J = 8.6 Hz, 1H), 8.17 (s, 1H), 8.10 (d, J = 8.6 Hz, 1H, NH), 8.09–8.02 (m, 3H), 7.91 (dd, J = 8.6 Hz, J = 8.2 Hz, 1H), 7.52–7.47 (m, 2H), 4.55 (m, 1H, CH), 3.83–3.46 (m, 20H, CH_2_), 3.25 (s, 6H, CH_3_). HRMS (ESI) *m*/*z*: 655.2454 calculated for C_35_H_39_N_2_NaO_7_S^+^ [M + Na]^+^, found 655.2407.

**N^C2.** 119.9 mg, yield 78%. ^1^H NMR ((CD_3_)_2_CO, 400 MHz, δ): 9.46 (s, 1H), 9.20 (d, J = 8.0 Hz, 1H, NH), 9.09 (d, J = 8.4 Hz, 1H), 8.84 (d, J = 8.4 Hz, 1H), 8.36 (d, J = 9.0 Hz, 1H), 8.20 (d, J = 8.6 Hz, 1H), 8.18 (s, 1H), 8.09–8.03 (m, 3H), 8.02 (d, J = 7.4 Hz, 1H, NH), 7.91 (dd, J = 8.6 Hz, J = 8.2 Hz, 1H), 7.87 (d, J = 8.2 Hz, 1H, NH), 7.52–7.47 (m, 2H), 5.15 (m, 1H, CH), 4.26 (m, 2H, CH), 3.65–3.38 (m, 40H, CH_2_), 3.33–3.25 (m, 12H, CH_3_). HRMS (ESI) *m*/*z*: 1047.4613 calculated for C_52_H_71_N_4_NaO_15_S^+^ [M + Na]^+^, found 1047.4636.

Synthesis of 4-(2-(pyridin-2-yl)-1H-phenanthro [9,10-d]imidazol-1-yl)benzoic acid. Phenanthrene-9,10-dione (100 mg, 0.480 mmol), 4-aminobenzoic acid (165 mg, 1.200 mmol), 2-pyridinebenzaldehyde (52 mg, 0.480 mmol), ammonium acetate (185 mg, 2.400 mmol), and 5 mL of glacial acetic acid were stirred at 70 °C in a 25 mL round-bottom flask. After 3 h, the solution was evaporated. While cooling to room temperature, the resulted oil turned to beige precipitate. It was washed and centrifuged three times with 2 mL of methanol. Then the product was recrystallized from hot acetone and finally dried under vacuum. Yield: 50 mg, 25%. ^1^H NMR ((CD_3_)_2_SO, 400 MHz, δ): 13.26 (br s, 1H), 8.95 (d, J = 8.3 Hz, 1H), 8.89 (d, J = 8.4 Hz, 1H), 8.72 (d, J = 7.9, 1H), 8.29 (d, J = 7.7 Hz, 2H), 8.17 (d, J = 8.5 Hz, 2H), 7.95 (dd, J = 8.3 Hz, J = 7.9 Hz, 1H), 7.81 (dd, J = 8.5, J = 8.1, 1H), 7.77 (d, J = 8.5 Hz, 2H), 7.72 (dd, J = 8.6, J = 8.0, 1H), 7.59 (dd, J = 8.4, J = 7.8, 1H), 7.38 (dd, J = 8.7, J = 8.3, 1H), 7.34 (dd, J = 8.8, J = 8.0, 1H), 7.05 (d, J = 8.6, 1H). HRMS (ESI) *m*/*z*: 438.1218 calculated for C_27_H_17_N_3_NaO_2_^+^ [M + Na]^+^, found 438.1197.

Synthesis of **N^N#** ligands (**N^N1** and **N^N2**). 4-(2-(pyridin-2-yl)-1H-phenanthro [9,10-d]imidazol-1-yl)benzoic acid (20.8 mg, 0.050 mmol), the corresponding amine (**NH_2_-2OEG** or **NH_2_-4OEG**, 0.055 mmol), benzotriazolyloxytripyrrolidinophosphonium hexafluorophosphate (PyBOP, 26.0 mg, 0.050 mmol), NEt_3_ (ca. 10 mg, 0.099 mmol), and dry DMSO (0.5 mL) were placed in a 5 mL vial. The reaction mixture was stirred for 18 h at RT and thoroughly evaporated. The dried residue was dissolved in 50 mL of chloroform and sequentially washed with (i) 5 mL of brine with 0.25 g of citric acid monohydrate, (ii) 5 mL of brine with 0.1 g of NaHCO_3_, and (iii) with 5 mL of pure brine. The organic layer was dried over Na_2_SO_4_ and then evaporated to give the oily substance. It was dissolved in EtOAc/n-hexane 5:1 mixture, and purified using column chromatography (sequentially eluted with EtOAc and EtOAc/methanol 5:1 mixture). After evaporation the resulting ligand was obtained as colorless viscous oil.

**N^N1.** 31.1 mg, yield 90%. ^1^H NMR (CD_3_OD, 400 MHz, δ): 8.86 (d, J = 8.0 Hz, 1H), 8.81 (d, J = 8.4 Hz, 1H), 8.76 (d, J = 8.0, 1H), 8.56 (d, J = 8.4 Hz, 1H, NH), 8.40 (d, J = 5.0 Hz, 1H), 8.07 (d, J = 8.4 Hz, 2H), 8.04 (d, J = 8.4 Hz, 1H), 7.89 (dd, J = 8.6, J = 8.0, 1H), 7.76 (dd, J = 8.4, J = 7.8, 1H), 7.72–7.66 (m, 3H), 7.57 (dd, J = 8.7, J = 8.3, 1H), 7.36 (dd, J = 8.8, J = 8.0, 1H), 7.28 (dd, J = 8.4, J = 7.8, 1H), 7.20 (d, J = 8.6, 1H), 4.49 (m, 1H, CH), 3.78–3.52 (m, 20H, CH_2_), 3.39–3.33 (m, 6H, CH_3_). HRMS (ESI) *m*/*z*: 715.3108 calculated for C_40_H_44_N_4_NaO_7_^+^ [M + Na]^+^, found 715.3089.

**N^N2.** 47.3 mg, yield 87%. ^1^H NMR (CD_3_OD, 400 MHz, δ): 8.87 (d, J = 8.1 Hz, 1H), 8.81 (d, J = 8.5 Hz, 1H), 8.77 (d, J = 8.1, 1H), 8.42 (d, J = 4.6 Hz, 1H), 8.11 (d, J = 8.6 Hz, 2H), 8.06–7.98 (m, 1H), 7.90 (dd, J = 8.5, J = 8.0, 1H), 7.76 (dd, J = 8.5, J = 8.0, 1H), 7.73–7.68 (m, 3H), 7.58 (dd, J = 8.6, J = 8.2, 1H), 7.38 (dd, J = 8.2, J = 7.8, 1H), 7.28 (dd, J = 8.4, J = 7.8, 1H), 7.18 (d, J = 9.2, 1H), 5.02 (m, 1H, CH), 4.21 (m, 2H, CH), 3.68–3.46 (m, 40H, CH_2_), 3.39–3.32 (m, 12H, CH_3_), 2.92–2.78 (m, 2H, CH_2_). HRMS (ESI) *m*/*z*: 565.2577 calculated for C_57_H_76_N_6_Na_2_O_15_^+^ [M + 2Na]^2+^, found 565.2566.

General procedure for the synthesis of dimeric complexes [Ir_2_(**N^C#**)_4_Cl_2_] (**D1**, **D2**). IrCl_3_·6H_2_O (12.2 mg, 0.030 mmol), the corresponding cyclometallating ligand (**N^C1** or **N^C2**, 0.063 mmol), NaHCO_3_ (4.8 mg, 0.057 mmol), 2-methoxyethanol (4.5 mL), and distilled water (1.5 mL) were placed in a 25 mL round-bottom flask. The reaction mixture was stirred at 100 °C for 18 h. The resulting dark-red solution was dried. The solid residue was dissolved in 1.5 mL of benzene and centrifuged. The resulting solution was dried, dissolved in CH_2_Cl_2_/n-hexane 5:1 mixture, and purified using column chromatography (sequentially eluted with CH_2_Cl_2_ and CH_2_Cl_2_/methanol 10:1 mixture). The solution obtained was evaporated, the dried residue was dissolved in 0.2 mL of acetone and then precipitated by adding 9.0 mL of Et_2_O. The precipitate was isolated by centrifugation and then it was thoroughly vacuum-dried to give the resulting dimeric complex.

[Ir_2_(**N^C1**)_4_Cl_2_] (**D1**). Dark-red solid, 36.6 mg, yield 82%.^1^H NMR (CD_3_OD, 400 MHz, 298 K, δ): 8.86 (d, J = 8.1 Hz, 1H), 8.78 (s, 1H), 8.68 (d, J = 8.1 Hz, 1H), 8.14 (d, J = 8.1 Hz, 1H), 7.91 (dd, J = 6.0 Hz, J = 7.0 Hz, 1H), 7.82 (dd, J = 6.0 Hz, J = 7.0 Hz, 1H), 7.34 (br d, 1H), 6.65 (br dd, 1H), 6.35 (d, J = 8.6 Hz, 1H), 5.99 (dd, J = 7.6 Hz, 7.2 Hz, 1H), 5.34 (d, J = 8.0 Hz, 1H), 4.43 (br m, 1H, CH), 3.88–3.38 (m, 20H, CH_2_), 3.31–3.19 (m, 6H, CH_3_). HRMS (ESI) *m*/*z*: 728.2328 calculated for C_70_H_79_IrN_4_O_14_S_2_^2+^ [Ir(**N^C1**)_2_+H]^2+^, found 728.2358.

[Ir_2_(**N^C2**)_4_Cl_2_] (**D2**). Dark-red solid, 53.2 mg, yield 78%.^1^H NMR (CD_3_OD, 400 MHz, 323K, δ): 8.95 (s, 1H), 8.94 (d, J = 8.6 Hz, 1H), 8.91 (d, J = 7.2 Hz, 1H), 8.81 (s, 1H), 8.62 (d, J = 8.3 Hz, 1H), 8.48 (d, J = 8.6 Hz, 1H), 8.03–7.90 (m, 5H), 7.85 (d, J = 8.3 Hz, 1H), 7.48 (d, J = 7.2 Hz, 2H), 6.74 (dd, J = 8.6 Hz, 8.2 Hz, 1H), 6.33 (d, J = 8.5 Hz, 1H), 6.24 (d, J = 9.0 Hz, 1H), 6.03–5.98 (m, 2H), 5.38 (m, 1H, CH), 5.34 (d, J = 8.4 Hz, 1H), 5.13 (m, 1H, CH), 4.52 (m, 1H, CH), 4.34 (m, 1H, CH), 4.25 (m, 2H, CH), 3.81–3.42 (m, 80H, CH_2_), 3.40–3.24 (m, 24H, CH_3_), 3.01–2.87 (m, 4H, CH_2_). HRMS (ESI) *m*/*z*: 747.6359 calculated for C_104_H_144_IrN_8_O_30_S_2_^3+^ [Ir(**N^C2**)_2_+2H]^3+^, found 747.6369.

General procedure for the synthesis of iridium [Ir(**N^C#**)_2_(**N^N#**)]Cl complexes (**Ir1**, **Ir2**). The corresponding Ir(III) dimeric complex (**D1** or **D2**, 0.010 mmol), the diimine ligand (**N^N1** or **N^N2**, 0.021 mmol) and acetone (2 mL) were placed in a 5 mL vial. The reaction mixture was stirred for 18 h at RT. The solution obtained was evaporated, the residue was dissolved in 1.0 mL of water and centrifuged. Then the water solution was thoroughly evaporated, the dried substance was dissolved in 0.2 mL of acetone and precipitated by adding 9.0 mL of Et_2_O. The resulting suspension was solicited and centrifuged. The solid residue was additionally two times washed with 10 mL of Et_2_O using sonication and centrifuged. The precipitate was thoroughly vacuum-dried to give dark-red solid of the resulting complex.

[Ir(**N^C1**)_2_(**N^N1**)]Cl (**Ir1**). 40.1 mg, yield 92%. ^1^H NMR (CD_3_OD, 400 MHz, 323K, δ): 9.50-9.43 (m, 2H), 9.35 (s, 1H), 9.28 (bd, 1H), 9.19 (d, J = 8.6 Hz, 1H), 9.10 (bd, 1H), 8.89 (d, J = 8.2 Hz, 1H), 8.76 (s, 1H), 8.60 (d, J = 7.6 Hz, 1H), 8.22-8.17 (m, 6H), 8.11-7.98 (m, 3H), 7.78 (bd, 1H), 7.68 (d, J = 8.0 Hz, 1H), 7.59-7.53 (m, 2H), 7.42 (s, 2H), 7.25 (dd, J = 9.0 Hz, J = 8.6 Hz, 2H), 7.19 (dd, J = 8.0 Hz, J = 8.6 Hz, 2H), 6.91 (d, J = 7.8 Hz, 1H), 6.75-6.64 (m, 3H), 5.17-4.90 (m, 2H, CH_2_), 4.38 (m, 2H, CH), 4.12 (m, 1H, CH), 3.62-3.50 (m, 60H, CH_2_), 3.32-3.14 (m, 18H, CH_3_). HRMS (ESI) *m*/*z*: 1085.8859 calculated for C_110_H_122_IrN_8_NaO_21_S_2_^2+^ [M + Na]^2+^, found 1085.8824.

[Ir(**N^C2**)_2_(**N^N2**)]Cl (**Ir2**). 60.0 mg, yield 89%. ^1^H NMR (CD_3_OD, 400 MHz, 323K, δ): 9.79 (d, J = 8.0 Hz, 1H), 9.72 (d, J = 8.0 Hz, 1H), 9.28 (s, 1H), 9.14 (d, J = 8.6 Hz, 1H), 9.00 (d, J = 8.4 Hz, 1H), 8.76 (s, 1H), 8.72 (d, J = 7.6 Hz, 1H), 8.67 (d, J = 8.6 Hz, 1H), 8.31-8.22 (m, 4H), 8.09 (d, J = 8.4 Hz, 1H), 8.01 (d, J = 8.2 Hz, 1H), 7.91 (d, J = 8.2 Hz, 1H), 7.81 (d, J = 8.3 Hz, 2H), 7.72 (d, J = 8.1 Hz, 2H), 7.66-7.59 (m, 3H), 7.53 (dd, J = 8.3 Hz, J = 8.1 Hz 1H), 7.42-7.30 (m, 3H), 7.25 (dd, J = 7.8 Hz, J = 8.0 Hz 1H), 7.21-7.14 (m, 3H), 6.76 (dd, J = 7.9 Hz, J = 8.3 Hz 1H), 6.72 (dd, J = 7.9 Hz, J = 8.3 Hz 1H), 6.65 (d, J = 8.4 Hz, 1H), 6.62-6.54 (m, 3H), 6.49 (dd, J = 7.7 Hz, J = 8.0 Hz 1H), 5.46 (m, 1H), 5.01 (m, 2H, CH), 4.87 (m, 1H, CH), 4.20 (m, 4H, CH), 4.09 (m, 2H, CH), 3.66-3.42 (m, 120H, CH_2_), 3.40-3.26 (m, 36H, CH_3_), 2.93-2.61 (m, 6H, CH_2_). HRMS (ESI) *m*/*z*: 1123.8030 calculated for C_161_H_218_IrN_14_Na_2_O_45_S_2_^2+^ [M + 2Na]^3+^, found 1123.8099.

General procedure for the synthesis of iridium [Ir(**N^C#**)_2_(**N^N#**)]PF_6_ complexes (**Ir1a**, **Ir2a**). The corresponding Ir(III) complex (**Ir1** or **Ir2**, 0.009 mmol), KPF_6_ (165.6 mg, 0.900 mmol) and acetone (2 mL) were placed in a 5 mL vial. The reaction mixture was stirred for 18 h at RT. The solution obtained was evaporated, suspended in 1.0 mL of 1,2-dichloroethane and centrifuged. The solid residue was sequentially washed two times with 0.3 mL of 1,2-dichloroethane and centrifuged. Then the combined 1,2-dichloroethane solution was evaporated. The dried substance was dissolved in 0.1 mL of acetone and precipitated by adding 4.5 mL of Et_2_O. The resulting suspension was solicited and centrifuged. The precipitate was thoroughly vacuum-dried to give dark-red solid of the resulting complex.

[Ir(**N^C1**)_2_(**N^N1**)]PF_6_ (**Ir1a**). 19.9 mg, yield 96%. Spectroscopic ^1^H, ^1^H-^1^H COSY and NOESY NMR (Appendix A) data are almost identical to those for the related **Ir1** complex with Cl^−^ counterion. HRMS (ESI) *m*/*z*: 1074.3936 calculated for C_110_H_123_IrN_8_O_21_S_2_^2+^ [M + H]^2+^, found 1074.3933.

[Ir(**N^C2**)_2_(**N^N2**)]PF_6_ (**Ir2a**). 29.0 mg, yield 93%. Spectroscopic ^1^H, ^1^H-^1^H COSY and NOESY NMR (Appendix A) data are almost identical to those for the related **Ir2** complex with Cl^−^ counterion. HRMS (ESI) *m*/*z*: 1109.1484 calculated for C_161_H_220_IrN_14_O_45_S_2_^2+^ [M + 2H]^3+^, found 1109.1521.

### 4.2. Photophysical Experiments

Photophysical measurements in solution were performed in aqueous media and, partially, in methanol. Absorption spectra were measured with a Shimadzu UV-1800 spectrophotometer (Shimadzu Corporation, Kyoto, Japan). The excitation spectra in solution were recorded using a Fluorolog-3 spectrofluorimeter (HORIBA Jobin Yvon Ltd., Kyoto, Japan). The emission spectra were registered using Avantes AvaSpec-2048 × 64 spectrometer (Avantes, Apeldoorn, Netherlands). The absolute emission quantum yield was determined in solution by a comparative method. LED (365 nm) was applied for pumping and [Ru(bpy)_3_][PF_6_]_2_ water solution (Φ = 0.040 air-saturated, 0.063 Ar-saturated) was used as the reference. Pulse laser TECH-263 Basic (wavelength 527 nm, pulse width 5 ns, repetition frequency 1000 Hz) (Laser Export, Moscow, Russia), a Hamamatsu H10682-01 photon counting head (Hamamatsu, Hamamatsu, Japan), a FASTComTec MCS6A1T4 multiple-event time digitizer (FAST ComTec, Oberhaching, Germany) and an Ocean Optics monochromator Monoscan-2000 (interval of wavelengths 1 nm) (Ocean Optics, Largo, FL, USA) were used for lifetime measurements. An oxygen meter PyroScience FireStingO2 (PyroScience GmbH, Aachen, Germany), equipped with an oxygen probe OXROB10 and a temperature sensor TDIP15, was used to determine partial pressure and concentration of molecular oxygen in aqueous solutions. Temperature control was performed by using a Quantum Northwest qpod-2e (Quantum Northwest Inc., Liberty Lake, WA, USA) cuvette sample compartment.

### 4.3. Cell Cultures

HeLa cells were maintained in Dulbecco’s modified Eagle medium (DMEM, Gibco, Carlsbad, CA, USA) with 10% fetal bovine serum (FBS, Biowest, Nuaille, France) and 1% penicillin/streptomycin (Gibco, Carlsbad, CA, USA). The Chinese hamster ovary CHO-K1 cells were cultured in DMEM/F12 (Biolot, St. Petersburg, Russia) medium supplemented with 10% FBS (Gibco, Carlsbad, CA, USA), 2 mM glutamine (Gibco, Carlsbad, CA, USA), and penicillin/streptomycin at a concentration 100 U/ mL (Thermo Fisher Scientific, Waltham, MA, USA). The murine colon carcinoma CT26 cells were cultured in DMEM (Gibco, Carlsbad, CA, USA) supplemented with 10% FBS (Gibco, Carlsbad, CA, USA), 2 mM glutamine (Gibco, Carlsbad, CA, USA), 10 µg/mL penicillin (Gibco, Carlsbad, CA, USA), and 10 mg/mL streptomycin (Gibco, Carlsbad, CA, USA).

All cell lines were maintained in a humidified incubator at 37 °C with 5% CO_2_ and passaged routinely using trypsin-EDTA (Thermo Fisher Scientific, Waltham, MA, USA). For live-cell confocal microscopy, the cells (1 × 10^5^ CT26 cells, 1 × 10^5^ HeLa cells, or 1 × 10^5^ CHO-K1 cells in 1.5–2 mL DMEM) were seeded in glass-bottomed 35 mm dishes (Ibidi GmbH, Gräfelfing, Germany) or Petri dishes with glass coverslips (Nunc, Thermo Fisher Scientific, Waltham, MA, USA) and incubated for 24–48 h until reaching a confluence of ~70%. **Ir2** and **Ir2a** complexes dissolved in water were added to the cells in a final concentration of 40 µM for CHO-K1 cells, 75 µM for HeLa cells, and 100 µM for CT26 cells. For the study of internalization dynamics, the cells were incubated with the complexes and imaged at different time-points starting from 20 min. Before the cell imaging, the medium in the glass bottom dishes was changed to DMEM without phenol red.

For PLIM experiments, the cells were incubated with **Ir2** and **Ir2a** for 3 h for CT26 cells and 24 h for HeLa and CHO-K1 cells and washed with fresh media prior to microscopy. Hypoxia was induced by placing a cover glass above the cell monolayer for 2 h.

### 4.4. MTT Assay

The cells (2 × 10^3^ CT26 cells, 2 × 10^4^ HeLa cells, 1 × 10^4^ CHO-K1 cells in 100–200 µL of culture medium/well) were seeded in 96-well plates (Nunc, Thermo Fisher Scientific, Waltham, MA, USA) and incubated overnight. The complexes were added to the cells at concentrations 0–150 µM for 24 h, afterwards the cells were treated with the MTT reagent 3(4,5-dimethyl-2-thiasolyl)-2,5-diphenyl-2H-tetrasole bromide (Thermo Fisher Scientific, Waltham, MA, USA) at the concentration 0.5 mg/mL according to the manufacturer’s protocol. Following further incubation at 37 °C under 5% CO_2_ for 4 h, the formazan crystals were dissolved in DMSO (Merck, Munich, Germany) and the absorbance was measured at 570 nm using a Thermo Scientific Multiskan FC microplate reader (Thermo Fisher Scientific, Waltham, MA, USA). The percentage of viable cells relative to the control was determined for each well. For each concentration of the complex MTT was repeated 30 times for CT26, 15 times for HeLa, and 5 times for CHO-K1 cells.

### 4.5. Cell Compartment Staining

For the vital staining of lysosomes and late endosomes in HeLa and CHO-K1 cells, LysoTracker Green DND-26 (Thermo Fisher Scientific, Waltham, MA, USA) was used at the concentration of 50 nM. For the vital staining of mitochondria in HeLa and CHO-K1 cells, MitoTracker Green FM (Thermo Fisher Scientific, Waltham, MA, USA) was used at the concentration of 50 nM. LysoTracker or MitoTracker was added into the culture medium for 30 min prior to confocal imaging. For the vital staining of the nuclei in HeLa and CHO-K1 cells, Hoechst 33342 (Thermo Fisher Scientific, Waltham, MA, USA) was used at the concentration of 1.6 μM for 5 min. For the staining of lysosomes and late endosomes in CT26 cells, LysoTracker Yellow HCK-123 (Thermo Fisher Scientific, Waltham, MA, USA) was used at the concentration of 3 µM. Mitochondria were identified by endogenous fluorescence from the reduced nicotinamide adenine dinucleotide NADH. The nuclei were stained with DAPI (Thermo Fisher Scientific, Waltham, MA, USA, 0.3 µM).

### 4.6. Confocal Microscopy and Colocalization Assay

The differential interference contrast (DIC) and fluorescence images of HeLa cells were taken using an Olympus FV3000 laser scanning confocal microscope (Olympus Corporation, Tokyo, Japan). Oil immersion lens 40/1.42× was used. The luminescence of **Ir2**, and **Ir2a** complexes was excited by a laser with a wavelength of 405 nm and recorded in the region of 690–790 nm. Hoechst 33342 fluorescence was excited at 405 nm and recorded in the 415–465 nm range. LysoTracker Green and MitoTracker Green fluorescence was excited at 488 nm and recorded in the 505–570 nm range.

Single sections or Z-series with a step of 1 μm were recorded (10–15 consecutive optical sections per cell). All data were obtained from at least three independent experiments. In each experiment, 6–8 fields containing 60–90 cells totally were imaged for each case.

Live CHO-K1 cells were imaged by using confocal inverted Nikon Eclipse Ti2 microscope (Nikon Corporation, Tokyo, Japan) with x60 oil immersion objective. Emission of iridium complexes was recorded using the excitation at 488 nm, emission at 663–738 nm. Hoechst 33342 fluorescence was excited at 405 nm and recorded in the 425–475 nm range. LysoTracker Green and MitoTracker Green fluorescence was excited at 488 nm and recorded in the 500–550 nm range. In addition to luminescent microphotographs, DIC images were also obtained.

The images were processed and analyzed using ImageJ software (National Institutes of Health, Bethesda, MY, USA). Z-stacks were projected onto single images using the max intensity method, then the analysis of **Ir** complexes’ luminescence intensity in cells was carried out using ImageJ.

The quantitative co-localization analysis was performed using ImageJ JACoP Plugin to determine Manders’ co-localization coefficient M1, which is defined as the sum of the intensities of the selected red objects containing green signal, divided by the sum of the intensities of all selected red objects. Thresholds were set by a visually estimated value for each channel. Results are represented as mean ± standard deviation.

Imaging of CT26 cells was performed on a laser scanning microscope LSM880 (Carl Zeiss, Jena, Germany). Oil immersion objective lens C Plan-Apochromat 40×/1.3 NA was used for image acquisition. Luminescence of **Ir2** and **Ir2a** was excited with a He-Ne laser at the wavelength of 543 nm, the emission was detected in the range 650–750 nm. Fluorescence of LysoTracker Orange was excited at the wavelength of 488 nm and detected in the range 500–600 nm. Fluorescence of NADH was excited in two-photon mode with a Ti:Sa femtosecond laser MaiTai HP (Spectra-Physics Inc., Milpitas, CA, USA) at 750 nm and detected in the range 450–490 nm. DAPI was exited at 405 nm, the emission was recorded in the 415–465 nm range. The luminescence images were analyzed using ImageJ software. The Manders’ co-localization coefficient M1 was assessed using ZEN software (Carl Zeiss, Jena, Germany).

Quantification of luminescence intensity of **Ir2** and **Ir2a** was performed in the cell cytoplasm by manual selection of the cytoplasm region of interest. Calculations were made for 30–50 cells in total from 5 fields of view for every time point. Dead cells identified by round morphology on the bright-field images were excluded from the analysis. Results are presented as mean ± standard deviation.

### 4.7. PLIM

Phosphorescence lifetime imaging microscopy (PLIM) of CHO-K1 and HeLa cells was carried out using the time-correlated single photon counting (TCSPC) DCS-120 module (Becker&Hickl GmbH, Berlin, Germany) integrated to the Nikon Eclipse Ti2 confocal devise. Picosecond laser (405 nm) was used as excitation source, 720/60 nm band pass filter was used to cut off background signal. PLIM images were obtained using the following settings: frame time 7.299 s, pixel dwell time 27.30 µs, time range of PLIM recording 16.40 µs, total acquisition time 120–130 s, and image size 512 × 512 pixels. Oil immersion 60× objective with zoom 5.33 provided a scan area of 0.05 mm × 0.05 mm. 

PLIM of CT26 was performed using LSM880 equipped with TCSPC module SPC-150 (Becker & Hickl GmbH, Berlin, Germany). Phosphorescence was excited in two-photon mode with a Ti:Sa femtosecond laser MaiTai HP with 80 MHz and 140 fs at 760 nm, the emission was detected in the “window” 596–660 nm. Images were collected with an oil immersion objective lens C Plan-Apochromat 40×/1.3 NA. Scanning was performed at a frame time of 17.275 s, corresponding to a pixel dwell time 65.9 µs, time range of PLIM recording was 19.20 µs, the total acquisition time was 150 s. Image size was 512 × 512 pixels.

Phosphorescence lifetime distribution was calculated using SPCImage 8.1 software (Becker & Hickl GmbH, Berlin, Germany). The phosphorescence decay curves were fitted in monoexponential decay mode with an average goodness of the fit 0.8 ≤ χ^2^ ≤ 1.2. The average number of photons per curve were not less than 5000 at binning 3–4. The data are presented as mean phosphorescence lifetime per cell.

### 4.8. Computational Details

All calculations were performed using the Gaussian 16computer code (Gaussian Inc., Wallingford, CT, USA) [63] in DFT methodology. A hybrid density functional with empirical correction of dispersion interactions, APFD [64], was used. For carbon and hydrogen atoms, the Pople’s 6-31G* basis set was chosen. The Pople’s 6-311+G* basis set was used for all other atoms except for iridium. The Stuttgart–Dresden effective core pseudopotential and the corresponding basis set were used for iridium [65]. The Polarizable Continuum Model (PCM) was applied to account for non-specific solvation [66].

To simplify the calculations, a complete geometry optimization (Figure 2) of the model complex with -C(O)NH-Me fragments in the N^C and N^N ligands was carried out instead of -C(O)NH-OEG moieties.

## 5. Conclusions

We synthesized and characterized three novel luminescent iridium complexes (**Ir1**, **Ir2**, and **Ir2a**), which contain the same chromophoric center shielded from interaction with environment by oligoethyleneglycole crowns of various branching, two of them (**Ir2** and **Ir2a**) are associated with different counterions. Photophysical study of these complexes in water, phosphate buffer (PBS) and in model biological media (FBS) revealed that these emitters display phosphorescence in NIR region with the quantum yields of ca. 12% in degassed aqueous solution and lifetime in microsecond domain. The complexes’ emissions display bi-exponential decay, the shorter component of which (τ_1_, contribution 10–20%) is independent of oxygen concentration, whereas the longer one (τ_2_) shows appreciable dependence on oxygen concentration to give linear Stern–Volmer plots that point to dynamic oxygen quenching of the complexes’ phosphorescence. The values ofτ_2_ obtained in different degassed media (water, PBS and FBS) coincide under the limit of experimental uncertainty that is indicative of effective protection of the chromophore from interaction with proteins and other biomolecules presented in FBS. This emission parameter is also independent of pH, probe concentration, and shows only slight dependence on media viscosity. All these observations indicate that these complexes are prospective for application as oxygen sensors in aqueous media by using the long (τ_2_) component of emission decay. However, the MTT study of the complexes showed that **Ir1** displays unacceptable toxicity in the concentration range used in bioimaging experiments that is most probably a result of weaker iridium center shielding by less branched OEG corona. The **Ir2** and **Ir2a** probes are considerably less toxic and show preferential localization in acidic endolysosomes upon incubation in HeLa, CT26, and CHO-K1 cells. The PLIM experiments with these complexes were successfully performed to analyze oxygen distribution in the cells under normoxia and hypoxia. These obtained results clearly demonstrated that the lifetime values of the calibration plots obtained in FBS media may be used for reliable estimation of oxygen concentration in cells. The probe’s localization in lysosomes paves the way to analysis of oxygen concentration changes in single endosomal structures, which is very important for understanding the mechanisms of redox reactions’ coordination in the whole cell during various intracellular processes. It is also worth noting that the PLIM lifetime maps obtained under aerated conditions show a certain inhomogeneity across the cell. However, the assignment of these lifetime variations requires additional studies, which are now in progress.

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
