# Peer review of "Biocompatible Ir(III) Complexes as Oxygen Sensors for Phosphorescence Lifetime Imaging"

_molecules, 2021, doi:10.3390/molecules26102898_

Round 1

Reviewer 1 Report

As there was no access to supplementary materials, the paper was poorly reviewed because there was no source data on which the authors based the conclusions written in the manuscript. 

The following question was raised during the reading of the paper:

... All complexes are luminescent in aqueous media to give emission in NIR region with a rather high quantum yield, which amounts up to ca. 12% in degassed solution... - how was calculated the percentage increase in emissions ?

Author Response

It is a pity that the file with supplementary materials because of technical reasons (compressed  ZIP file containing all necessary information has been provided during submition) was not available to reviewers. Supplementary materials have been now added to the revised version of the manuscript after the main text and bibliography (to exclude a possibility of another loss of this file). Please accept my apologies for this shortcoming.

The phrase «… quantum yield, which amounts up to ca. 12% ...» may sound ambiguous, therefore, in the revised version of the article, we changed it for «… quantum yield, up to ca. 12% ...», since with these words we were going to give the absolute value of the quantum yield in deaerated solution, but not its percentage change upon degasation.

Reviewer 2 Report

Biocompatible Ir(III) Complexes as Oxygen Sensors for Phosphorescence Lifetime Imaging by Ilya S. Kritchenkov et al. is an original research article where the authors propose new compounds for detection of oxygen concentration in living cells. The article presents an interesting solution, especially concerning the engineering of the probes. In short, the problem of the study is presented in the introduction. The synthetic solution is invented, which is a valuable contribution to chemistry of inert and luminescent iridium coordination compounds. Lastly, viability of the idea has been verified on cell lines. Altogether, the article is a concise and well designed interdisciplinary study, which is recommended for publication after minor corrections.

  1. The supplementary information lacks in the mass spectra for each compound. The authors are asked to provide the spectra for evaluation.

  2. The wording, especially the phrase bio compatible seem to be overused throughout the text, and in fact it is not stressed why the advantage of bio compatibility is so important and what in fact it is. The cells are in fact biological structures, which live, so constant underlining this state of matter seem to be an exaggeration to some point. The same counts for detailed, which seems to be exaggerated (when it comes to NMR) but in fact the word detailed matches microscopy (rather than NMR or ESI MS).

  1. The supplementary information lacks in the mass spectra for each compound. The authors are asked to provide the spectra for evaluation.
  2. The wording, especially the phrase biocompatibile seem to be overused thrughot the text, and in fact it is not stressed why the advantage of biocompatibility is so important and what in fact it is.

Author Response

It is a pity that the file with supplementary materials because of technical reasons (compressed  ZIP file containing all necessary information has been provided to editorial office) was not available to reviewers. Supplementary materials have been now added to the revised version of the manuscript after the main text and bibliography (to exclude a possibility of another loss of this file). Please accept my apologies for this shortcoming.

The Supplementary materials containing ESI MS and NMR spectra, photophysical and biological details as well as other experimental data are now available in the revised version of the manuscript.

We agree with the overusing of the words “biocompatible” and “detailed” and revised the manuscript accordingly.

Reviewer 3 Report

This is an interesting and thorough paper describing the synthesis and characterisation of iridium complexes for PLIM. It is good to see both the synthesis and cellular studies combined in a single paper as it highlights in the case of Ir1 that while it may be synthesised and studied chemically there are often issues such as cellular toxicity that impact on the biology. This does not detract from the study.

The introduction is a little long and strays into reviewing other work ( rather thna just providing context) and includes background that should be known to practitioners in this field

Author Response

We agree with this remark and revised the introduction by removing some background information. Nevertheless, we think that comparison of the previous studies (called as “reviewing other work”) with that described in the manuscript deserved to be presented in introduction.
